# Baroreceptor Sensitivity Predicts Functional Outcome and Complications after Acute Ischemic Stroke

**DOI:** 10.3390/jcm8030300

**Published:** 2019-03-03

**Authors:** Ching-Huang Lin, Cheng-Chung Yen, Yi-Ting Hsu, Hsin-Hung Chen, Pei-Wen Cheng, Ching-Jiunn Tseng, Yuk-Keung Lo, Julie Y.H. Chan

**Affiliations:** 1Department of Biological Sciences, National Sun Yet-Sen University, Kaohsiung 80424, Taiwan; chlin2524@vghks.gov.tw (C.-H.L.); lavender4999@gmail.com (Y.-T.H.); 2Section of Neurology, Kaohsiung Veterans General Hospital, Kaohsiung 81362, Taiwan; ccyen@vghks.gov.tw (C.-C.Y.); yklo@vghks.gov.tw (Y.-K.L.); 3Department of Physical Therapy, Shu-Zen Junior College of Medicine and Management, Kaohsiung 82144, Taiwan; 4Department of Medical Education and Research, Kaohsiung Veterans General Hospital, Kaohsiung 81362, Taiwan; shchen0910@gmail.com (H.-H.C.); peiwen420@gmail.com (P.-W.C.); cjtseng@vghks.gov.tw (C.-J.T.); 5Institute for Translational Research in Biomedicine, Kaohsiung Chang Gung Memorial Hospital, Kaohsiung 83301, Taiwan

**Keywords:** acute stroke, baroreflexes, intubation, prognostic factors

## Abstract

Autonomic dysfunctions including impaired baroreflex sensitivity (BRS) can develop after acute ischemic stroke (AIS) and may predispose patients to subsequent cardiovascular adverse events and serve as potential indicators of long-term mortality. This study aimed to determine the potential short-term prognostic significance of BRS after AIS. All patients admitted to Kaohsiung Veterans General Hospital within 72 h after onset of first-ever AIS between April 2008 and December 2012 were enrolled. Autonomic evaluation with continuous 10-minute monitoring of beat-to-beat hemodynamic and intracranial parameters was performed within 1 week after stroke by using the Task Force Monitor and transcranial Doppler. The 176 enrolled AIS patients were divided into high-BRS and low-BRS groups. All but two enrolled patients (who died within 3 months after stroke) attended scheduled follow-ups. The high-BRS group had significantly lower National Institutes of Health Stroke Scale (NIHSS) scores at 1 and 2 weeks after stroke and at discharge; lower modified Rankin scale (mRS) scores 1, 3, 6, and 12 months after stroke; and lower rates of complications and stroke recurrence compared to the low-BRS group. This study provides novel evidence of the utility of BRS to independently predict outcomes after AIS. Furthermore, modifying BRS may hold potential in future applications as a novel therapeutic strategy for acute stroke.

## 1. Introduction

Clinical studies have demonstrated that acute stroke can alter autonomic function [1,2]. Post-stroke autonomic dysfunction presents as sympathovagal imbalance with high sympathetic and low parasympathetic activities, manifested as impaired baroreflex sensitivity (BRS). Impaired BRS has been observed in acute stroke patients compared to control subjects matched for age, sex, and blood pressure (BP) [1]. 

Impaired autoregulation and BP fluctuation after stroke may significantly alter cerebral perfusion [3] and adversely influence outcome [4]. Impaired BRS is independently associated with acute hypertension [5], elevated risk of arrhythmia and mortality after acute stroke [6]. Impaired BRS has a long-term independently prognostic value, as revealed by a follow-up study over four years after acute ischemic infarction (AIS) [1]. A significantly association between impaired BRS and short-term outcome was also observed in patients with acute intracerebral hemorrhage, but this association has remained unclear in AIS [7]. 

Acute stroke is associated with increased susceptibility to infections, most frequently of respiratory and urinary tracts, which can adversely affect mortality and outcome [8]. Aspiration and dysphagia alone cannot fully explain the high infection rate [9]. Several studies have suggested that systemic infection is associated with post-stroke symptomatic hyperactivity [10]. BRS has been identified as an independent predictor of early infection in patients with intracerebral hemorrhage, but not in AIS [11]. Impaired autoregulation [12], temporarily elevated BP [13], cardiac events [14] and increased susceptibility to infections [8,11] occurring after AIS may worsen patient outcome and increase mortality. This study aimed to investigate the prognostic value of BRS for short-term functional outcome, complications (infection), and mortality in patients with AIS.

## 2. Materials and Methods

### 2.1. Subjects

All patients admitted to the stroke unit of Kaohsiung Veterans General Hospital within 72 h after neuroradiologically confirmed first-ever AIS stroke onset from 2008 to 2012 were enrolled in this prospective study (Figure 1). Patients were included in the analysis only if they fulfilled all of the following criteria: (1) admission for first-ever AIS; (2) computed tomography (CT) or magnetic resonance imaging (MRI) evidence of acute ischemic lesion consistent with clinical manifestations verified by a neurologist; (3) absence of clinically relevant arrhythmia, including atrial fibrillation, on admission; (4) absence of any major concurrent diseases, including uremia, malignancies, chronic obstructive pulmonary disease, acute myocardial infarction, unstable angina, or other conditions involving autonomic dysfunction; (5) absence of fever, alterations of consciousness, or any relevant hemodynamic compromise on admission; and (6) absence of hypotensive agents needed to stabilize the patient’s condition within 1 week after stroke onset.

### 2.2. Standard Protocol Approvals, Registrations, and Patient Consents

The procedures were approved by an ethical standards committee on human experimentation (Kaohsiung Veterans General Hospital, VGHKS97-CT1-01, VGHKS98-CT3-05, VGHKS10-CT10-21). All patients or their next of kin gave written informed consent to participate.

### 2.3. Protocol

Upon admission, each enrolled patient underwent clinical, neurological, radiological, autonomic, sonographic and functional examinations (Figure 1). Autonomic function assessment with simultaneous intracranial Doppler examination was performed within 1 week after stroke onset. During the hospitalization period for acute stroke, patients were monitored for complications of pneumonia and urinary tract infection (UTI). After discharge, all enrolled patients were regularly followed for 12 months to evaluate neurological functional outcome, mortality, and recurrent stroke events. 

### 2.4. Radiological Examinations

Brain CT was performed in each patient on arrival at the emergency room (ER) to exclude the possibility of intracerebral hemorrhage or previous stroke. Patients also underwent brain MRI to evaluate the presence of acute ischemic infarction. 

### 2.5. Carotid Duplex Ultrasonography 

Extracranial and intracranial carotid duplex ultrasonography was performed in each subject within 1 week after stroke onset. Extracranial artery stenosis was diagnosed when color-coded duplex ultrasound (PHILIPS iEe33) with a 7.5-MHz probe for extracranial examination revealed extracranial carotid artery stenosis ≥50%. Intracranial artery stenosis was diagnosed as turbulent or disturbed flow in the intracranial carotid, vertebral, or basilar arteries on transcranial Doppler sonography (SPENCER TECHNOLOGIES 3, Model PMD 150). Intimal–medial thickness (IMT) over the bilateral common carotid arteries was measured by B-mode ultrasound. 

### 2.6. Clinical BRS Examinations

Clinical BRS function testing was carried out in each patient within 7 days after stroke onset. All measurements were performed in a specialized humidity- and temperature-controlled laboratory. The use of nicotine, alcohol, caffeine, anticholinergic, and BP-lowering agents was not permitted before the tests. The BRS was assessed using the Task Force Monitor 3040i system (CNSystems Medizintechnik GmbH, Graz, Austria). The patients were subjected to noninvasive continuous recording of arterial BP and beat-to-beat heart rate from a “Flying-V” finger cuff for a minimum of 30 min to obtain baseline measurements. The value of the recorded BP was periodically verified by the measurements from the oscillometric BP cuff placed on the patient’s upper arm. For assessment of the BRS, each patient was measured in resting supine position for 10 min, tilted to 70° for another 10 min, and returned to resting supine position for the last 10 min. BRS was analyzed using a validated technique involving synchronization of systolic blood pressure (SBP) and rate–rate (RR) interval data. The α-index of BRS was calculated with the same low-frequency and high-frequency bands using the cross-spectral densities between RR and SBP variability. The assessment of BRS have improved in recent decades and “non-invasive” quantification of BRS might become highly relevant in the clinical setting, being applicable to wide populations [15]. The head-up tilted method was used to perform autonomic function testing instead of deep breathing and Valsalva maneuver methods because stroke patients cannot readily cooperate with the latter methods. 

### 2.7. Neurological Functional Assessments

Neurologic severity in each subject was evaluated with the National Institutes of Health Stroke Scale (NIHSS) at admission, at 1 and 2 weeks after stroke, and at discharge. Post-stroke functional outcome was evaluated with the modified Rankin scale (mRS) and Barthel Index (BI) at discharge and at 1, 3, 6, and 12 months after stroke onset by the same neurologist. 

### 2.8. Complications During Hospitalization

During the hospitalization period for AIS, episodes of post-stroke pneumonia and UTI, defined by Centers for Disease Control and Prevention (CDC)/National Healthcare Safety Network (NHSN) surveillance definition of health care-associated infection and criteria in 2008 [16], of the enrolled patients were recorded. Pneumonia was defined as: (1) ≥2 serial chest radiographs with at least one of the following: new or progressive and persistent infiltrate, consolidation, or cavitation; (2) at least one of the following: (i) fever (>38 °C) with no other recognized cause, (ii) leukopenia (<4000 WBC/mm^3^) or leukocytosis (>12,000 WBC/mm^3^), (iii) for adults ≥70 years old, altered mental status with no other recognized cause; and (3) at least two of the following: (i) new onset of purulent sputum or change in character of sputum or increased respiratory secretions or increased suctioning requirements, (ii) new onset or worsening cough, or dyspnea, or tachypnea, (iii) rales or bronchial breath sounds, and (iv) worsening gas exchange, increased oxygen requirements, or increased ventilator demand). UTI was defined as at least one of the following criteria: (1) at least one of the following signs or symptoms with no other recognized cause: fever (>38 °C ), urgency, frequency, dysuria, or suprapubic tenderness and a positive urine culture, that is, 105 microorganisms per cc of urine with no more than two species of microorganisms; and (2) at least two of the following signs or symptoms with no other recognized cause: fever (>38 °C), urgency, frequency, dysuria, or suprapubic tenderness, and at least one of the following: (i) positive dipstick for leukocyte esterase and/or nitrate, (ii) pyuria, (iii) organisms seen on Gram’s stain of unspun urine, (iv) at least 2 urine cultures with repeated isolation of the same uropathogen, (v) physician diagnosis of a urinary tract infection, or (vi) physician institutes appropriate therapy for a urinary tract infection. The events of pneumonia and UTI were recorded as complications. The presence of complications was verified by the attending neurologist for each patient. 

### 2.9. Hematology and Serum Biochemical Analyses

Upon admission, fasting blood samples were collected from each patient to assess their complete blood count, prothrombin time, and levels of blood creatinine, total cholesterol, high-density lipoprotein, low-density lipoprotein, triglycerides, and glutamic pyruvate transaminase.

### 2.10. Mortality/Recurrent Stroke

Mortality and recurrent stroke events during the follow-up period were verified by the attending neurologist.

### 2.11. Data Availability

Anonymized data can be obtained by request from any qualified investigator for purposes of replicating procedures and results.

### 2.12. Statistics

All statistical values and categorical variables are presented as mean ± standard deviation and percentage, respectively. Initial comparisons were made with the Student’s *t*-test, one-way analysis of variance, chi-square, or Fisher’s exact test as appropriate. Newman–Keuls post-hoc comparison was used to assess significant main effects within groups for the one-way analysis of variance. Univariate analysis of variance was used to explore the influence of single factors on outcome measures. Significant variables in the univariate analysis were then analyzed with a multivariate logistic regression model to evaluate the relationship between predictive variables and outcome. A stepwise regression analysis was employed to select the variables into a regression model to predict the outcome. All statistical analyses were performed using SPSS software version 22.0. The threshold for statistical significance was set at *p* < 0.05.

## 3. Results

One hundred and seventy-six AIS patients (age: 62.9 ± 12.3 years; 135 males, 41 females) were enrolled in this study between April 2008 and December 2012 and divided into a high-BRS group (≥9 ms/mmHg) and low-BRS group (<9 ms/mmHg). The baseline characteristics of all patients are shown in Table 1. The high-BRS group had a significantly higher rate of smoking and significantly lower diabetes prevalence, intimal thickness, C-reactive protein, and glycosylated hemoglobin (HbA1c) levels compared to the low-BRS group. There was no significant difference in age, sex, body mass index (BMI), hypertension, family history of stroke, alcohol, hypercholesterolemia, or hypertriglyceridemia between the groups. The percentage of patients in each group who received thrombolytic therapy with recombinant tissue plasminogen activator (rtPA) was also similar.

Table 2 shows the neurological and functional outcomes overall and in the low- and high-BRS groups. At initial ER assessment, there was no significant difference in NIHSS score between the groups. However, the high-BRS group had a significantly lower NIHSS score at 1 week and 2 weeks after stroke and at discharge compared to the low-BRS group (Figure 2A). 

The high-BRS group had significantly lower mRS scores at 1, 3, 6, and 12 months after stroke compared to the low-BRS group. The high-BRS group also had a higher BI at initial admission, discharge, 1 month, 3 months, 6 months, and 12 months after stroke (Figure 2B). In terms of morbidity and mortality, the high-BRS group had lower rates of complications (pneumonia and UTI), recurrent stroke, and events (recurrent stroke or death). The rate of hemorrhagic transformation did not significantly differ between the groups.

We then divided the enrolled patients according to functional outcome at 1 month after stroke into independent (mRS = 0–2) and dependent (mRS = 3–6) groups. Table 3 shows the baseline characteristics of these groups at 1 month after stroke. The independent group was younger, with lower rates of hypertension and diabetes and higher rates of smoking and BRS within 1 week of admission ≥ 9. There was no significant difference in sex, heart disease, family history of stroke, alcohol consumption, hypercholesterolemia, hypertriglyceridemia, or the rate of rtPA infusion.

Table 4 shows the results of multiple logistic regression analysis for mRS at 1 month after stroke, and complications during hospitalization for AIS. The independent predictors of independence (mRS 0–2) at 1 month after stroke were BRS within 1 week, NIHSS in the ER, and age. The independent predictors of complications during acute hospitalization were BRS within 1 week and NIHSS in the ER. Hypertension, diabetes, smoking, and rtPA therapy were not independent predictors of independence (mRS 0–2) at 1 month after stroke, or complications during acute hospitalization. Age was not an independent predictor of complications during acute hospitalization. 

## 4. Discussion

This study investigated the prognostic value of BRS for patient mortality and secondary events up to 12 months after AIS. Previous research has shown that BRS, a critical regulator of arterial pressure, is an independent predictor of long-term mortality after AIS. This study reports for the first time that BRS is an independent predictor for short-term functional outcome (modified Rankin scale at 1 month after stroke) and complications (pneumonia, UTI) during hospitalization.

Post-stroke autonomic dysfunction increases the risk of BP variability, cardiac arrhythmia, myocardial damage, increased platelet aggregation, coronary vasoconstriction, and mortality [1,17]. The arterial baroreceptor reflex is a key neural mechanism involved in short-term regulation of the cardiovascular system [18]. BRS has been investigated as a predictor of hypertension, coronary artery disease, myocardial infarction, chronic heart failure, and stroke outcomes [1,7,17,19].

Robinson et al. [1] found that impaired BRS was associated with increased long-term mortality (28% vs. 8%) over follow-up for more than four years after AIS independent of other variables including age, BP, stroke subtype, and stroke severity. However, cardiac BRS did not significantly differ between patients with mRS ≥3 or mRS <3 at 1 month after AIS (*p* = 0.21). In one study, BRS was not an independent predictor of unfavorable outcome at 3 months after AIS [20], while in another it was found to be an independent predictor of patient outcome, as measured by NIHSS, mRS, and the Glasgow Coma Scale, at day 10 after acute intracerebral hemorrhage, but not AIS [7]. Our study is the first to identify BRS as an independent predictor of functional outcome in terms of independence (mRS = 0–2) or dependence (mRS = 3–6) at 1 month after AIS. Notably, BRS was not an independent predictor of independence (mRS = 0–2) or dependence (mRS = 3–6) at 3 months, 6 months, and 1 year after stroke (Appendix A). Possible reasons are not immediate clear but may include drug compliance, motivation in rehabilitation, lifestyle, and medical services received in the days following hospital discharge.

Medical complications are common after stroke and may extend the length of hospital stay, worsen functional outcome, and increase cost of care [21]. A meta-analysis of 87 studies including a total of 137,817 stroke patients found a pooled overall infection rate of 30%, with pneumonia and UTI each occurring in 10% of patients [22]. Pneumonia is the most common infection after stroke and associated with a relative mortality risk of 3.0 in a study of 14,293 stroke patients [23]. A systematic review found that dysphagia occurred in 37–78% of stroke patients and increased the risk for pneumonia 3-fold overall and 11-fold in patients with confirmed aspiration [24]. However, nearly half of patients who develop pneumonia after stroke do not experience aspiration, indicating that other mechanisms such as post-stroke immunosuppression are involved. UTI occurs frequently after stroke and is associated with poorer neurological functional status [25]. Compared to the general population, stroke patients, whether catheterized or not, are more susceptible to UTIs during hospitalization. In this study we found that in contrast to no incidence in AIS patients with high BRS, 10% of the patients with low BRS developed pneumonia and UTI during hospitalization (cf. Table 2). These data are comparable to the reports in the literature and indicate BRS as an independent predictor for complications of pneumonia and UTI during hospitalization. The possible reasons for this increased risk of UTI include increased bladder dysfunction, increased likelihood of bladder catheterization, and immunosuppression [26]. 

Acute stroke may lead to immunosuppression, which is in turn related to susceptibility to infection [8]. Experimental and clinical evidence suggests that sympathetic hyperactivity is associated with pro-inflammatory cytokine production [27], decreased lymphocyte activation, shift from Th1 to Th2 cytokine predominance [28] and stroke-related immunosuppression syndrome [10]. Post-stroke sympathetic hyperactivity and decreased BRS with subsequent immunosuppression has been associated with the occurrence of infections after stroke and implicated in secondary brain injury [11]. In the present study, the high-BRS group had a significantly lower rate of pneumonia and UTI during hospitalization for AIS than the low-BRS group. Moreover, the present study is the first to report that BRS within 1 week after stroke independently predicts the occurrence of infection during AIS hospitalization. 

Odemuyiwa et al. [29] investigated the influence of thrombolytic therapy on the evolution of BRS after myocardial infarction (MI) and found that mean BRS at day 6 after MI was higher in patients treated with thrombolytic agents than in untreated patients, but BRS was similar between the groups at 6 weeks and 3 months after MI. In the present study, the rates of intravenous thrombolytic therapy did not differ between patients with acute stroke in the high- and low-BRS groups. 

The BRS over Heart Rate Variability (HRV) was used in this study since the method specifically addressed the theme of this study. On the other hand, HRV is a generalized assessment of autonomic function, based on the balance between sympathetic and parasympathetic regulation of the heart [30]. The present study has some limitations worth noting. First, the study was performed at a single institution. Second, we excluded patients with arrhythmia, uremia, malignancy, chronic obstructive pulmonary disease, acute myocardial infarction, unstable angina, consciousness change, or unstable hemodynamic condition. Third, the total sample size was limited to 176 patients. Fourth, breathing rate, which may have a significant impact on BRS [31], was not controlled during the test due to the clinical condition of the patients were relatively unstable. Therefore, the results cannot be applied to all patients with ischemic stroke. Further studies should be conducted to address these limitations and confirm the significance of BRS in patients with AIS.

## 5. Conclusions

This study demonstrated that BRS within 1 week after stroke is an independent predictor of functional outcome (dependence) at 1 month after stroke and complications during hospitalization for AIS. These findings suggest that assessing BRS can provide valuable information regarding prognosis and clinical management, particularly as complications during hospitalization for acute stroke may worsen clinical outcome. However, the pathophysiologic mechanisms underlying the observed associations are not yet established. In the future, modifying autonomic dysfunction after acute stroke may be considered as a novel therapeutic strategy to improve functional outcomes and decrease morbidity in patients with AIS.

## Figures and Tables

**Figure 1 jcm-08-00300-f001:**
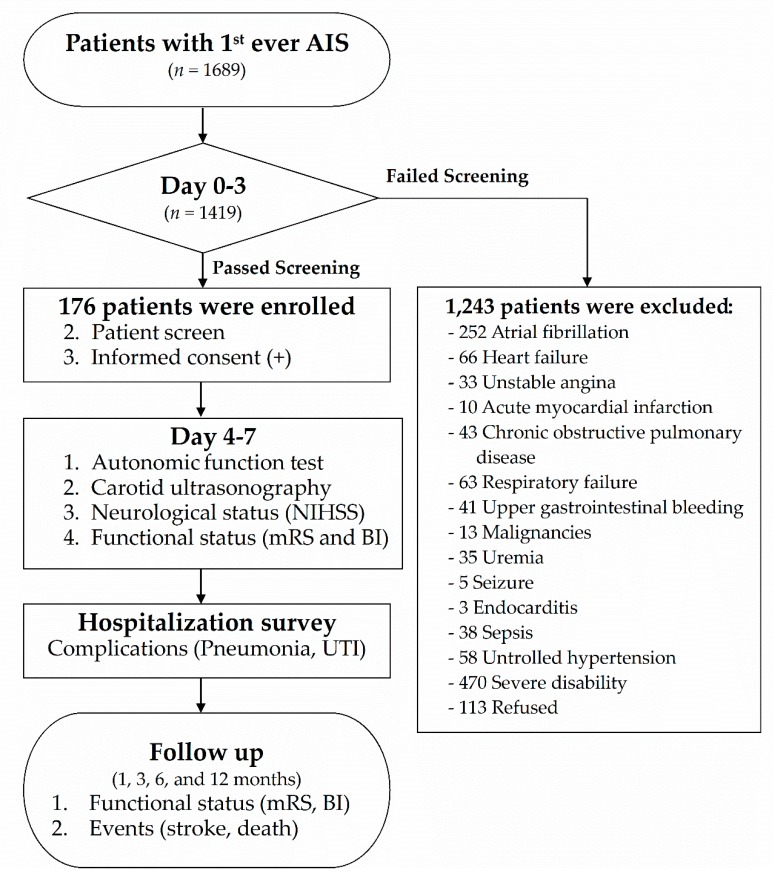
Flow chart of the study. AIS: acute ischemic stroke; UTI: urinary tract infection; BI: Barthel Index; NIHSS: National Institutes of Health Stroke Scale; mRS: modified Rankin scale.

**Figure 2 jcm-08-00300-f002:**
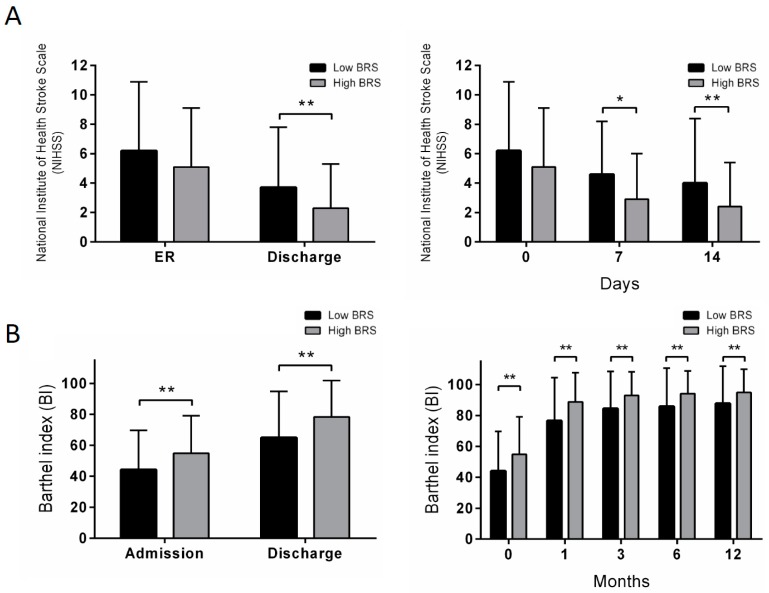
Neurologic severity and functional outcome in low-BRS and high-BRS groups at different time points after acute ischemic stroke. (**A**) comparison of NIHSS score in low-BRS and high-BRS groups at ER and discharge, and post-stroke 7 and 14 days; (**B**) comparison of BI of low-BRS and high-BRS groups at admission, discharge, post-stroke 1, 3, 6 and 12 months. BRS: baroreflex sensitivity measured within 1 week after stroke; ER, emergency room. * *p* < 0.05 vs. the low-BRS group. ** *p* < 0.01 vs. the low-BRS group.

**Table 1 jcm-08-00300-t001:** Baseline characteristics of all patients and low- and high-BRS groups.

Demographic Variables	Total (*n* = 176)	Low-BRS (*n* = 99)	High-BRS (*n* = 77)	*p*-Value
Age (years, mean)	62.9 ± 12.3	64.4 ± 11.2	61.1 ± 13.3	0.08
Sex (male, %)	135 (76.7%)	74 (74.7%)	61 (79.2%)	0.49
BMI (mean ± SD)	24.8 ± 3.5	24.6 ± 3.7	25.0 ± 3.2	0.51
Hypertension (*n*, %)	149 (84.7%)	90 (90.9%)	59 (76.6%)	0.09
Diabetes (*n*, %)	74 (42.0%)	51 (51.5%)	23 (29.9%)	0.004
Family history of stroke	63 (39.4%)	38 (42.2%)	25 (35.7%)	0.403
Smoking (*n*, %)	87 (49.4%)	42 (42.4%)	45 (58.4%)	0.035
Alcohol (*n*, %)	34 (19.3%)	18 (18.2%)	16 (20.8%)	0.665
Hypercholesterolemia (*n*, %)	141 (80.6%)	81 (81.8%)	60 (78.9%)	0.634
Hypertriglyceridemia (*n*, %)	84 (47.4%)	44 (44.9%)	37 (50.7%)	0.453
CCA-IMT (mm)	6.2 ± 1.2	6.4 ± 1.2	5.9 ± 1.2	0.026
TOAST (*n*, %)				0.012
Atherothrombotic	66 (37.5%)	39 (39.4%)	27 (35.1%)	
Lacunar	91 (51.7%)	55 (55.6%)	36 (46.8%)	
Cardiac embolism	7 (4.0%)	0 (0%)	7 (9.1%)	
Other determined	0 (0%)	0 (0%)	0 (0%)	
Undetermined	12 (6.8%)	5 (5.1%)	7 (9.1%)	
Ant/post circulation (*n*, %)				0.08
Anterior	124 (70.5%)	75 (75.8%)	49 (63.6%)	
Posterior	52 (29.5)	24 (24.2%)	28 (36.4%)	
CRP	1.1 ± 2.2	1.5 ± 2.8	0.5 ± 0.8	0.028
HbA1c (%)	7.8 ± 2.4	8.4 ± 2.6	7.0 ± 1.8	0.004
rtPA infusion (*n*, %)	28 (15.9%)	17 (17.2%)	11 (14.3%)	0.604

BRS: baroreflex sensitivity measured within 1 week after stroke; BMI: body mass index; CCA: common carotid artery; IMT: intima–media thickness; TOAST: Trial of ORG 10172 in Acute Stroke Treatment classification; CRP: C-reactive protein; HbA1c: glycosylated hemoglobin; rtPA: recombinant tissue plasminogen activator. Low BRS: BRS < 9 ms/mmHg; High BRS: BRS ≥ 9 ms/mmHg.

**Table 2 jcm-08-00300-t002:** Neurological and functional outcomes in all patients and low- and high-BRS groups.

Demographic Variables	Total (*n* = 176)	Low-BRS (*n* = 99)	High-BRS (*n* = 77)	*p*-Value
NIHSS, in ER	5.7 ± 4.4	6.2 ± 4.7	5.1 ± 4.0	0.089
NIHSS, 7 days	3.8 ± 3.5	4.6 ± 3.6	2.9 ± 3.1	0.044
NIHSS, 14 days	3.3 ± 3.9	4.0 ± 4.4	2.4 ± 3.0	0.004
NIHSS, discharge	3.1 ± 3.7	3.7 ± 4.1	2.3 ± 3.0	0.008
mRS, 1 month	2.7 ± 1.1	2.9 ± 1.2	2.3 ± 1.0	0.001
mRS, 3 months	2.1 ± 1.3	2.3 ± 1.4	1.8 ± 1.2	0.006
mRS, 6 months	1.7 ± 1.5	2.0 ± 1.5	1.4 ± 1.3	0.003
mRS, 12 months	1.3 ± 1.5	1.6 ± 1.6	1.0 ± 1.4	0.005
BI, admission	48.8 ± 25.5	44.1 ± 25.6	54.9 ± 24.2	0.005
BI, discharge	70.7 ± 28.1	64.8 ± 30.0	78.3 ± 23.5	0.001
BI, 1 month	81.9 ± 25.1	76.6 ± 27.9	88.8 ± 18.9	0.001
BI, 3 months	88.2 ± 21.0	84.5 ± 24.0	92.9 ± 15.4	0.006
BI, 6 months	89.5 ± 21.4	85.8 ± 24.9	94.2 ± 14.6	0.006
BI, 12 months	90.8 ± 21.0	87.7 ± 24.2	94.9 ± 15.1	0.017
Complications	20 (11.4%)	18 (18.2%)	2 (2.6%)	0.001
Pneumonia	9 (5.1%)	9 (9.1%)	0 (0%)	0.007
UTI	9 (5.1%)	9 (9.1%)	0 (0%)	0.001
Event	8 (4.5%)	8 (8.1%)	0 (0.0%)	0.01
Recurrent stroke	6 (3.4%)	6 (6.1%)	0 (0.0%)	0.036
Death	2 (1.1%)	2 (2%)	0 (0.0%)	0.210
Hemorrhagic transformation	5 (2.8%)	4 (4.0%)	1 (1.3%)	0.388

Data are presented as median ± standard deviation or *n* (%). BRS: baroreflex sensitivity measured within 1 week after stroke; NIHSS: National Institutes of Health Stroke Scale; ER, emergency room; mRS: modified Rankin scale; BI: Barthel index; UTI: urinary tract infection. Low BRS: BRS < 9 ms/mmHg; High BRS: BRS ≥ 9 ms/mmHg. Complications: pneumonia, urinary tract infection. Event: recurrent stroke + death.

**Table 3 jcm-08-00300-t003:** Baseline characteristics of modified Rankin Scale groups at 1 month.

Demographic Variables	mRS 0–2 (*n* = 77)	mRS 3–6 (*n* = 99)	*p*-Value
Age (years, mean)	59 ± 12.2	66 ± 11.5	<0.001
Sex (male, %)	61 (79.2%)	74 (74.7%)	0.590
Hypertension	60 (77.9%)	89 (89.9%)	0.035
Diabetes	24 (31.2%)	50 (50.5%)	0.014
Heart disease	7 (9.1%)	10 (10.1%)	1.000
Family history of stroke	27 (35.1%)	36 (36.4%)	0.745
Smoking	45 (58.4%)	42 (42.4%)	0.048
Alcohol	18 (23.4%)	16 (16.2%)	0.665
Hypercholesterolemia	62 (80.5%)	79 (79.8%)	0.848
Hypertriglyceridemia	36 (46.8%)	45 (45.5%)	0.877
BRS within 1 week ≥ 9	45 (58.4%)	32 (32.3%)	0.001
rtPA infusion	10 (13.0%)	18 (18.2%)	0.410

Data are presented as *n* (%) unless otherwise indicated. mRS: modified Rankin scale; BRS: baroreflex sensitivity measured within 1 week after stroke; rtPA: recombinant tissue plasminogen activator.

**Table 4 jcm-08-00300-t004:** Multiple logistic regression for dependence 1 month after stroke and complications during hospitalization.

Variables	mRS				Complications			
	OR	95% CI		*p*	OR	95% CI		*p*
BRS	2.11	1.02	4.36	0.044	7.60	1.53	37.85	0.013
NIHSS (ER)	1.36	1.18	1.56	<0.001	1.23	1.10	1.38	<0.001
Age	1.05	1.02	1.09	0.004	1.03	0.98	1.08	0.273
HTN	1.81	0.64	5.11	0.260	1.15	0.17	7.65	0.888
Diabetes	1.98	0.93	4.18	0.075	0.79	0.26	2.38	0.673
Smoking	0.70	0.34	1.44	0.330	0.98	0.32	2.94	0.964
IV tPA	0.38	0.12	1.19	0.095	1.02	0.27	3.79	0.977

OR: odds ratio; CI: confidence interval; mRS: modified Rankin scale; BRS: baroreflex sensitivity; NIHSS (ER): NIH Stroke Scale score at emergency room admission; HTN: hypertension; IV tPA: intravenous tissue plasminogen activator.

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
