# Peer review of "Baroreceptor Sensitivity Predicts Functional Outcome and Complications after Acute Ischemic Stroke"

_jcm, 2019, doi:10.3390/jcm8030300_

Reviewer 1 Report

The article is simple to read and clear.

There are a number of points that I would like the researchers to address in revising the document.

Key points:

1.      What is the advantage of BRS over HRV, which has been studied extensively in the context of autonomic injury and functional abilities after stroke. It is worth referring to this point.

a.      Researchers have information on HRV according to the instrumentation used (PPG?) Perhaps should be introduced.

b.      If the PPG method was used can the authors, please indicate its limitations?

2.      Breathing rate might have a huge impact on BRS.

a.      Have you control the breathing rate?

b.      Please refer to this point.

3.      Why is a high and low BRS threshold value selected in this work?

a.      Please refer to this point.

b.      Is the division into two groups optimal? Why not divide into 3 ms / mmHg intervals and display (if any) a "dose" effect?

4. Data analysis. For example, analysis of the data in Table 2 perpetuates a difference at the beginning of the follow-up, but in the analysis of repeated observations, it is appropriate that the gap between the groups is maintained over time so that the time effect will disappear from the result. No interaction appears to exist. The whole view is a bit misleading.

5. Since there are confunders at entry to follow up, perhaps multivariate analysis with neutralization of these variables is desirable?

Author Response

Responses to comments from Reviewer 1:

 We appreciate the comments of Reviewer 1 and believe that our manuscript has been improved by attention to him or her. The followings are our responses to the specific issues raised:

 All the revisions are in red-marked.

 1)   What is the advantage of BRS over HRV, which has been studied extensively in the context of autonomic injury and functional abilities after stroke. It is worth referring to this point.

a.      Researchers have information on HRV according to the instrumentation used (PPG?) Perhaps should be introduced.

b.      If the PPG method was used can the authors, please indicate its limitations?

Reply: Thank you for the valuable suggestion.

1. Heart rate variability (HRV) and baroreceptor sensitivity (BRS) are the established and reliable parameters to quantitate the activities of sympathetic and parasympathetic function of cardiovascular autonomic control after acute stroke[1]. HRV is the variation over time of the period between consecutive heartbeats (R-R interval). It reflects the balance between the sympathetic and the parasympathetic tone and their effect on the sinus node[2]. The baroreflex arc is important in the short-term regulation of the cardiovascular system and compensates for spontaneous fluctuations in blood pressure[3]. Central lesions, for example due to stroke, seem to decrease BRS, leading to a shift toward sympathetic dominance and blood pressure elevation or instability, which may negatively affect cerebral perfusion[4].

 2. Our team have no any idea to address BRS is better than HRV. We think that both parameters of HRV and BRS are important indicators and correlated disease severity. Our team chose BRS as our study target because of α-index of BRS, calculated with the same low-frequency and high-frequency bands using the cross-spectral densities between RR and SBP variability, may be a simpler value for clinicians to apply.

 3. We have no HRV information according to instrumentation used by photoplethysmography (PPG). Thanks for your great suggestion. This method may be discussed by our team in future study.

 4. We used the Task Force Monitor 3040i system (CNSystems Medizintechnik GmbH, Graz, Austria) to perform noninvasive continuous recording of arterial BP and beat-to-beat heart rate for a minimum of 30 minutes to obtain baseline measurements. Each patient was measured in resting supine position for 10 minutes, tilted to 70° for another 10 minutes, and returned to resting supine position for the last 10 minutes. BRS was automatically analyzed using a validated technique involving synchronization of SBP (systolic blood pressure) and RR (rate–rate) interval data. The α-index of BRS was calculated with the same low-frequency and high-frequency bands using the cross-spectral densities between RR and SBP variability. The head-up tilted method was used to perform autonomic function testing instead of deep breathing and Valsalva maneuver methods because stroke patients cannot readily cooperate with the latter methods.

 2)   Breathing rate might have a huge impact on BRS.

a.      Have you control the breathing rate?

b.      Please refer to this point.

 Reply: Thank you for this suggestion.

1. Stroke may disrupt breathing either by (A) causing a disturbance of central rhythm generation, (B) interrupting the descending respiratory pathways leading to a reduced respiratory drive, or (C) causing bulbar weakness leading to aspiration[5]. Respiration is commonly ventilated after stroke and the pattern of breathing may reflect the etiology, localization, and severity of the underlying cerebrovascular disease.

 2. Because the enrolled subjects were patients with first-ever acute ischemic stroke. Their clinical conditions were relatively unstable and were not eligible to cooperate our breath rate control.

 3) Why is a high and low BRS threshold value selected in this work?

a.      Please refer to this point.

b.      Is the division into two groups optimal? Why not divide into 3 ms / mmHg intervals and display (if any) a "dose" effect?

Reply: Thank you very much for your constructive suggestion.

1. We used the Task Force Monitor 3040i system (CNSystems Medizintechnik GmbH, Graz, Austria) to perform autonomic function test. The operation manual lists the reference of Task Force Monitor Vital Normal Values as below. The normal valve is defined as BRS>9.3 ms/mmHg. In our study, we divided all enrolled patient into a high-BRS group (≥9 ms/mmHg) and low-BRS group (<9 ms/mmHg).

2. Thanks for your constructive suggestion about “dividing into 3ms/mmHg intervals and display a dose effect”. This great point will be discussed in our future study.

 4) Data analysis. For example, analysis of the data in Table 2 perpetuates a difference at the beginning of the follow-up, but in the analysis of repeated observations, it is appropriate that the gap between the groups is maintained over time so that the time effect will disappear from the result. No interaction appears to exist. The whole view is a bit misleading.

Reply: Thank you for the valuable comments.

1. The correlation between BRS and long-term functional status remains unclear. Robinson et al. found no difference in BRS at30 days poststroke[6]. In contrast, Sykora et al. reported BRS to be an independent predictor of unfavorable outcome at three months after ischemic stroke, although a link with hyperglycemia might be present[7]. Although the supporting data are limited, HRV and BRS seem to correlate with stroke severity, early and late complications, and stroke outcome. Large-scale prospective studies applying internationally accepted standards of measures for analysis ofHRV and BRS are needed in patients with acute stroke[1].

 2. Table 2 showed a series change of neurological severity (NIHSS) and functional status (mRS, Barthel index) at post-stroke different time intervals among high-BRS and low BRS groups. The information of post-stroke neurological & functional outcomes at different time intervals showed by table 2 is more detailed than previous published studies. We want to show the whole story of post-stroke functional outcome among high-BRS and low-BRS groups. Readers may illustrate the detailed history of table 2 and organize a more comprehensive autonomic study in acute stroke.

 3. The value of BRS which was assessed within 7 days after acute stroke may tell us whole story. The high-BRS group demonstrated a significantly greater improvement in mRS between admission and discharge. This result shows the high rehabilitation potential in high-BRS group. The clear and definite correlations may need further study to verify.

 5) Since there are confunders at entry to follow up, perhaps multivariate analysis with neutralization of these variables is desirable?

Reply: Thank you for the valuable comments.

1. Univariate analysis of variance was used to explore the influence of single factors on outcome measures. Significant variables in the univariate analysis were then analyzed with a multivariate logistic regression model to evaluate the relationship between predictive variables and outcome. A stepwise regression analysis was employed to select the variables into a regression model to predict the outcome.

 2. The modified Rankin Scale (mRS) is a commonly used scale for measuring the degree of disability or dependence in the daily activities of people who have suffered a stroke or other causes of neurological disability. It has become the most widely used clinical outcome measure for stroke clinical trials[8,9]. We then divided the enrolled patients according to functional outcome at 1 month after stroke into independent (mRS=0–2) and dependent (mRS=3–6) groups.

 3. Table 3 shows the baseline characteristics of these groups at 1 month after stroke. The independent group was younger, with lower rates of hypertension and diabetes and higher rates of smoking and BRS within 1 week of admission ≥ 9. There was no significant difference in sex, heart disease, family history of stroke, alcohol consumption, hypercholesterolemia, hypertriglyceridemia, or the rate of rtPA infusion.

 4. Table 4 shows the results of multiple logistic regression analysis for mRS at 1 month after stroke, tube placement, and complications during hospitalization for AIS. NIHSS in the ER (initial stroke severity) and BRS (autonomic dysfunction parameters) remain the independent predictors of independence (mRS 0–2) at 1 month after stroke, tube placement and complications during acute hospitalization were BRS within 1 week and NIHSS in the ER. The independent predictors of complications during acute hospitalization after controlling other confounders.

  References:

1.         Yperzeele, L.; van Hooff, R.J.; Nagels, G.; De Smedt, A.; De Keyser, J.; Brouns, R. Heart rate variability and baroreceptor sensitivity in acute stroke: A systematic review. International journal of stroke : official journal of the International Stroke Society 2015, 10, 796-800.

2.         Hilz, M.J.; Moeller, S.; Akhundova, A.; Marthol, H.; Pauli, E.; De Fina, P.; Schwab, S. High nihss values predict impairment of cardiovascular autonomic control. Stroke 2011, 42, 1528-1533.

3.         Sykora, M.; Diedler, J.; Poli, S.; Rupp, A.; Turcani, P.; Steiner, T. Blood pressure course in acute stroke relates to baroreflex dysfunction. Cerebrovasc Dis 2010, 30, 172-179.

4.         Benarroch, E.E. The arterial baroreflex: Functional organization and involvement in neurologic disease. Neurology 2008, 71, 1733-1738.

5.         Howard, R.S.; Rudd, A.G.; Wolfe, C.D.; Williams, A.J. Pathophysiological and clinical aspects of breathing after stroke. Postgraduate medical journal 2001, 77, 700-702.

6.         Robinson, T.G.; Dawson, S.L.; Eames, P.J.; Panerai, R.B.; Potter, J.F. Cardiac baroreceptor sensitivity predicts long-term outcome after acute ischemic stroke. Stroke 2003, 34, 705-712.

7.         Sykora, M.; Diedler, J.; Poli, S.; Rizos, T.; Kellert, L.; Turcani, P.; Steiner, T. Association of non-diabetic hyperglycemia with autonomic shift in acute ischaemic stroke. Eur J Neurol 2012, 19, 84-90.

8.         Wilson, J.T.; Hareendran, A.; Grant, M.; Baird, T.; Schulz, U.G.; Muir, K.W.; Bone, I. Improving the assessment of outcomes in stroke: Use of a structured interview to assign grades on the modified rankin scale. Stroke 2002, 33, 2243-2246.

9.         Saver, J.L.; Filip, B.; Hamilton, S.; Yanes, A.; Craig, S.; Cho, M.; Conwit, R.; Starkman, S. Improving the reliability of stroke disability grading in clinical trials and clinical practice: The rankin focused assessment (rfa). Stroke 2010, 41, 992-995.

Reviewer 2 Report

In this manuscript, Ching-Huang and colleagues describe the association between altered baroreceptor sensitivity in patients with stroke and a higher incidence of complications and poorer outcomes, especially one month after the stroke. 

 The most exciting aspect of this study is that it suggests that autonomic dysfunction measured by the baroreflex sensitivity may be a contributor of complications and finally of the outcomes of patients with stroke. I find that the weakest part is the study of the association between baroreflex sensitivity and the placement of nasogastric tubes or Foley catheters. The authors include as an essential part of the study, but I think that in particular the placement of nasogastric tubes may be primarily explained by motor deficits causing dysphagia. In other words, it may reflect the severity of the deficits, and its association with altered baroreflex activity may, therefore, reflect baseline differences in patients in the severity of the stroke rather than reflecting a mechanistic relationship with autonomic dysfunction resulting in higher risk of dysphagia. Overall, I think that this part of the study does not add much and can be confusing. 

 Here I detail other concerns related to the different sections of the manuscript: 

-       Methods- Figure 1: I think that it would be interesting to include in the flow diagram the information on the number of patients that were not included in the study of each of the criteria mentioned in the text.

-       Methods – Neurological functional assessments: as explained in this section, the mRS was designed to evaluate functional outcomes after stroke, but it may not be very informative to describe the deficits at hospital admission. I would, therefore, suggest not giving this information in Table 2, and deleting the sentence in line 187 related to this result. 

-       Methods- Definition of pneumonia: the sentence explaining the operative definition of pneumonia is confusing. The authors should explain whether they used consensus criteria such as those published by Smith et al. (Stroke 2015) or not, and in any case, rephrase this part of the text because it is unclear which of the criteria were needed in order to diagnose this critical complication. 

-       Results/Discussion: as I mentioned above, it is hard to understand the point of measuring nasogastric tube or Foley catheter placements as an outcome measure related to the quality of baroreflex sensitivity. I suggest that the authors remove this part, or just mention it but also discuss the potential confounders that could be explaining these differences. 

-       Discussion: although most of the content of the discussion is correct, I think it is too long and that the authors could focus in the main findings of the study to make it more appealing for the reader. 

 Minor issues:

 -       Introduction: Line 45: … prognostic value over follow-up 1508 days after acute ischemic infarction. As day counts over 1000 are hard to relate to other time units, I would suggest reflecting the median follow-up of the study referenced in this sentence as “long follow-ups of over four years” or something similar. 

Author Response

Responses to comments from Reviewer 2:

 We appreciate the constructive comments of Reviewer 2 and believe that quality of the revised manuscript has improved by incorporating the comments by the Reviewer. The followings are our responses to the specific issues raised:

 All the revisions are marked in the revised manuscript in red-color.
1) Methods- Figure 1: I think that it would be interesting to include in the flow diagram the information on the number of patients that were not included in the study of each of the criteria mentioned in the text.

 Reply: Thank you for the valuable suggestion

1. We have added the information on the number of patients that were not included in the flow diagram. The revised new flow chart is shown below: (P. 3)

 2) Methods – Neurological functional assessments: as explained in this section, the mRS was designed to evaluate functional outcomes after stroke, but it may not be very informative to describe the deficits at hospital admission. I would, therefore, suggest not giving this information in Table 2, and deleting the sentence in line 187 related to this result. 

 Reply: Thank you for the valuable suggestion.

1. The modified Rankin Scale (mRS) is a commonly used scale for measuring the degree of disability or dependence in the daily activities of people who have suffered from stroke or other causes of neurological disability. It has become the most widely used clinical outcome measure for stroke clinical trials [1, 2].

 2. Table 2 shows a series changes in neurological severity (NIHSS) and functional status (mRS, Barthel index) at post-stroke different time intervals of both high and low BRS groups. The information of post-stroke neurological and functional outcomes at different time intervals showed in table 2 is more detailed than previous published studies. We trust that this will provide a more comprehensive information and benefit the readers with a greater appreciation of the significance of measuring BRS function in acute stroke patients.

 3. Although there was no significant difference in mRS score at admission between the two groups, the high-BRS group had demonstrated a significantly greater improvement in mRS between admission and discharge, in comparison to the low-BRS group (Figure 2B). High-BRS group has higher rehabilitation potential than low-BRS group during acute stroke hospitalization.

 4. Thanks for your constructive suggestion. If you don’t agree to our reasons of maintaining the mRS at hospital admission in table 2 and line 192 after revised manuscript, we will delete the content as your suggestion.

 3)  Methods- Definition of pneumonia: the sentence explaining the operative definition of pneumonia is confusing. The authors should explain whether they used consensus criteria such as those published by Smith et al. (Stroke 2015) or not, and in any case, rephrase this part of the text because it is unclear which of the criteria were needed in order to diagnose this critical complication. 

Reply: Thank you very much for your constructive suggestion.

We have revised the methods on the definition of stroke-associated pneumonia according consensus criteria published by Smith et al. (Stroke 2015) [3]. We have also added a new reference for it. “ Pneumonia was defined by consensus criteria (Smith et al.) [3]: (1) at least 1 of the following: (i)Fever (>38°C) with no other recognized cause, (ii)Leukopenia (<4000 WBC/mm3) or leukocytosis (>12 000 WBC/mm3), (iii)altered mental status with no other recognized cause for adults ≥70 y old; (2) at least 2 of the following: (i)new onset of purulent sputum, or change in character of sputum over a 24 h period, or increased suctioning requirements, (ii)new onset or worsening cough, or dyspnea, or tachypnea (respiratory rate>25/min), (iii)rales, crackles, or bronchial breath sounds, (iv)worsening gas exchange, increased oxygen requirements; (3)and ≥2 serial chest radiographs with at least 1 of the following: new or progressive and persistent infiltrate, consolidation, or cavitation. “ (P. 4, Line 122-130)

 4)  Results/Discussion: as I mentioned above, it is hard to understand the point of measuring nasogastric tube or Foley catheter placements as an outcome measure related to the quality of baroreflex sensitivity. I suggest that the authors remove this part, or just mention it but also discuss the potential confounders that could be explaining these differences. 

Reply: Thank you very much for your constructive suggestion.

1. Our study is the first to demonstrate that the correlations between BRS within 1 week after stroke and Tube (nasogastric tube and Foley) placement during acute stroke hospitalization. Indeed, there are potential confounders exist and are needed to verified by further comprehensive study. In the article, we discuss the issues of nasogastric tube & Foley and BRS and we hope this will that it can throw light on the future research.

(P. 10, Line 282-284, 292-293, 295-296)

If you don’t agree to our reasons of maintaining the issues of nasogastric tube & Foley and BRS after revised manuscript, we will delete the content as your suggestion.

  5) Discussion: although most of the content of the discussion is correct, I think it is too long and that the authors could focus in the main findings of the study to make it more appealing for the reader. 

Reply: Thank you for the valuable comments.

1. We have deleted some descriptions in Discussion per your suggestion.

 6) Introduction: Line 45: … prognostic value over follow-up 1508 days after acute ischemic infarction. As day counts over 1000 are hard to relate to other time units, I would suggest reflecting the median follow-up of the study referenced in this sentence as “long follow-ups of over four years” or something similar. 

Reply: Thank you for the kind comments.

1. We have changed the description in article from over follow-up 1508 days to long follow-ups of over four years.

(P. 2, Line 45) (P. 9, Line 242)

   Reference:

[1] J.T. Wilson, A. Hareendran, M. Grant, T. Baird, U.G. Schulz, K.W. Muir, I. Bone, Improving the assessment of outcomes in stroke: use of a structured interview to assign grades on the modified Rankin Scale, Stroke, 33 (2002) 2243-2246.

[2] J.L. Saver, B. Filip, S. Hamilton, A. Yanes, S. Craig, M. Cho, R. Conwit, S. Starkman, Improving the reliability of stroke disability grading in clinical trials and clinical practice: the Rankin Focused Assessment (RFA), Stroke, 41 (2010) 992-995.

[3] C.J. Smith, A.K. Kishore, A. Vail, A. Chamorro, J. Garau, S.J. Hopkins, M. Di Napoli, L. Kalra, P. Langhorne, J. Montaner, C. Roffe, A.G. Rudd, P.J. Tyrrell, D. van de Beek, M. Woodhead, A. Meisel, Diagnosis of Stroke-Associated Pneumonia: Recommendations From the Pneumonia in Stroke Consensus Group, Stroke, 46 (2015) 2335-2340.

Round  2

Reviewer 1 Report

Sorry, the authors try to response to my comments but he answers gave in this supplement file as well as in the manuscript do not cover the main topics that has been raised. 

1)   What is the advantage of BRS over HRV, which has been studied extensively in the context of autonomic injury and functional abilities after stroke. It is worth referring to this point.

a.      Researchers have information on HRV according to the instrumentation used (PPG?) Perhaps should be introduced.

b.      If the PPG method was used can the authors, please indicate its limitations?

Reply: Thank you for the valuable suggestion.

1.     Heart rate variability (HRV) and baroreceptor sensitivity (BRS) are the established and reliable parameters to quantitate the activities of sympathetic and parasympathetic function of cardiovascular autonomic control after acute stroke[1]. HRV is the variation over time of the period between consecutive heartbeats (R-R interval). It reflects the balance between the sympathetic and the parasympathetic tone and their effect on the sinus node[2]. The baroreflex arc is important in the short-term regulation of the cardiovascular system and compensates for spontaneous fluctuations in blood pressure[3]. Central lesions, for example due to stroke, seem to decrease BRS, leading to a shift toward sympathetic dominance and blood pressure elevation or instability, 

2. Our team have no any idea to address BRS is better than HRV. We think that both parameters of HRV and BRS are important indicators and correlated disease severity. Our team chose BRS as our study target because of α-index of BRS, calculated with the same low-frequency and high-frequency bands using the cross-spectral densities between RR and SBP variability, may be a simpler value for clinicians to apply. which may negatively affect cerebral perfusion[4].

So what is the added benefit in assessing BRS? Please expend this point in the text

2.     We have no HRV information according to instrumentation used by photoplethysmography (PPG). Thanks for your great suggestion. This method may be discussed by our team in future study.

This was the question; what was the technology (not the device) for assessing continuous recording of arterial BP. Was it PPG? If not, please present the thechnology. If it was based on PPG, please indicate its limitations.

 4. We used the Task Force Monitor 3040i system (CNSystems Medizintechnik GmbH, Graz, Austria) to perform noninvasive continuous recording of arterial BP and beat-to-beat heart rate for a minimum of 30 minutes to obtain baseline measurements. Each patient was measured in resting supine position for 10 minutes, tilted to 70° for another 10 minutes, and returned to resting supine position for the last 10 minutes. BRS was automatically analyzed using a validated technique involving synchronization of SBP (systolic blood pressure) and RR (rate–rate) interval data. The α-index of BRS was calculated with the same low-frequency and high-frequency bands using the cross-spectral densities between RR and SBP variability. The head-up tilted method was used to perform autonomic function testing instead of deep breathing and Valsalva maneuver methods because stroke patients cannot readily cooperate with the latter methods.

2)   Breathing rate might have a huge impact on BRS.

a.      Have you control the breathing rate?

b.      Please refer to this point.

 Reply: Thank you for this suggestion.

1. Stroke may disrupt breathing either by (A) causing a disturbance of central rhythm generation, (B) interrupting the descending respiratory pathways leading to a reduced respiratory drive, or (C) causing bulbar weakness leading to aspiration[5]. Respiration is commonly ventilated after stroke and the pattern of breathing may reflect the etiology, localization, and severity of the underlying cerebrovascular disease.

2. Because the enrolled subjects were patients with first-ever acute ischemic stroke. Their clinical conditions were relatively unstable and were not eligible to cooperate our breath rate control.

This is a limitation; please add this to the limitation section. 

3) Why is a high and low BRS threshold value selected in this work?

a.      Please refer to this point.

b.      Is the division into two groups optimal? Why not divide into 3 ms / mmHg intervals and display (if any) a "dose" effect?

Reply: Thank you very much for your constructive suggestion.

1. We used the Task Force Monitor 3040i system (CNSystems Medizintechnik GmbH, Graz, Austria) to perform autonomic function test. The operation manual lists the reference of Task Force Monitor Vital Normal Values as below. The normal valve is defined as BRS>9.3 ms/mmHg. In our study, we divided all enrolled patient into a high-BRS group (≥9 ms/mmHg) and low-BRS group (<9 ms/mmHg).

2. Thanks for your constructive suggestion about “dividing into 3ms/mmHg intervals and display a dose effect”. This great point will be discussed in our future study. Why in the future?

  4) Data analysis. For example, analysis of the data in Table 2 perpetuates a difference at the beginning of the follow-up, but in the analysis of repeated observations, it is appropriate that the gap between the groups maintained over time so that the time effect will disappear from the result. No interaction appears to exist. The whole view is a bit misleading.

Please change this table presentation.

Reply: Thank you for the valuable comments.

1. The correlation between BRS and long-term functional status remains unclear. Robinson et al. found no difference in BRS at30 days poststroke[6]. In contrast, Sykora et al. reported BRS to be an independent predictor of unfavorable outcome at three months after ischemic stroke, although a link with hyperglycemia might be present[7]. Although the supporting data are limited, HRV and BRS seem to correlate with stroke severity, early and late complications, and stroke outcome. Large-scale prospective studies applying internationally accepted standards of measures for analysis ofHRV and BRS are needed in patients with acute stroke[1].

 Author Response

Responses to comments from Reviewer 1:

 We appreciate the constructive comments by the Reviewer and believe that quality of the second revised manuscript has improved significantly by incorporating the suggestions by the Reviewer. The followings are our responses to the specific issues raised:

 All the revisions are marked in the revised manuscript in red-color.

1) What is the advantage of BRS over HRV, which has been studied extensively in the context of autonomic injury and functional abilities after stroke. It is worth referring to this point.

a. Researchers have information on HRV according to the instrumentation used (PPG?) Perhaps should be introduced.

b. If the PPG method was used can the authors, please indicate its limitations?

This was the question; what was the technology (not the device) for assessing continuous recording of arterial BP. Was it PPG? If not, please present the technology. If it was based on PPG, please indicate its limitations.

Response: Once again, we thank the Reviewer for the comment. Respectfully we submit that the method we used in this study specifically addressed baroreceptor sensitivity, the theme of this study. On the other hand, HRV is a generalized assessment of autonomic function, based on the balance between sympathetic and parasympathetic regulation of the heart. This narration now appears on p. 9, lines 277-279. In the present study we used the Task Force Monitor 3040i system (CNSystems Medizintechnik GmbH, Graz, Austria) to perform noninvasive continuous recording of arterial BP and beat-to-beat heart rate from a “Flying-V” finger cuff (p. 3, lines 101-1103)  

 2) Breathing rate might have a huge impact on BRS.

a. Have you control the breathing rate?

b. Please refer to this point.

This is a limitation; please add this to the limitation section.

Response: We thank the Reviewer for this comment. We are fully aware the impact of respiratory rhythm on BRS. Because the enrolled subjects were patients with first-ever acute ischemic stroke, their clinical conditions were relatively unstable and were not eligible to cooperate to our breath rate control. Consequently, we did not control the breathing rate. This limitation is now added in the revised manuscript. (p. 9, lines 283-285)

 3) Why is a high and low BRS threshold value selected in this work?

a. Please refer to this point.

b. Is the division into two groups optimal? Why not divide into 3 ms/mmHg intervals and display (if any) a "dose" effect?

Response: We thank the Reviewer for the suggestion. Respectfully we submit that in our original study the enrolled patients were divided into two groups, low and high BRS, based on normal BRS valve of BRS≧9.3 ms/mmHg defined by the Task Force Monitor 3040i system which we used to perform BRS test. Per suggestion, we have carried out addition analysis in which BRS were grouped by 3 ms/mmHg intervals. In the total 176 enrolled patients, 9 patients had BRS<3 mmhg="" and="" the="" patient="" numbers="" of="" brs="" between="" 18-21="">21 were also relative few (£20) to perform analysis. As a consequence, we divided the enrolled patients into 3 groups: BRS-L (BRS<6 ms/mmHg, n = 48), BRS-M (BRS between 6-9 ms/mmHg, n = 51) and BRS-H (BRS≧9 ms/mmHg, n = 77). (Table 1)

 Using this new grouping, we have performed multiple logistic regression analysis for dependence 1 month after stroke, as well as tube placement and complications during hospitalization among the 3 groups. The results showed that in comparison to BRS-H group the odds ratio (OR) was 1.82 (P=0.159) in BRS-M group and 2.58 (P=0.045) in BRS-L group to predict dependence 1 month after stroke (Table 2). Similar results were obtained with respect to the prediction on the rate of tube placement.

Moreover, the OR in BRS-M and BRS-L group to predict the rate of complications during hospitalization was 7.81 (P=0.02) and 7.42 (P=0.021), respectively, in comparison to the BRS-H group. Collectively, these results showed the BRS-L (BRS<6 ms/mmHg) and BRS-M (BRS between 6-9 ms/mmHg) shared similar features in predicting the rate of tube placement and complications during hospitalization, and no “dose” effect was found from the analysis. Therefore, we are confident with the original grouping of low and high BRS to predict the dependence at 1 month after stroke, the rate of tube placement, and the rate of complications during hospitalization.

 4) Data analysis. For example, analysis of the data in Table 2 perpetuates a difference at the beginning of the follow-up, but in the analysis of repeated observations, it is appropriate that the gap between the groups maintained over time so that the time effect will disappear from the result. No interaction appears to exist.

Please change this table presentation.

Response: Thank you for this valuable comment. Per suggestion by Reviewer #2, we have deleted the mRS score at admission, discharge and admission-discharge and Tube (nasogastric tube and Foley catheterization) in Table 2. The revised Table 2 shows a series changes in neurological severity (NIHSS) and functional status (mRS, Barthel index) at different post-stroke time intervals in high-BRS and low-BRS groups (p. 5-6).

 Once again, we thank the Reviewer for your insightful review and constructive comments to improve on the quality of the present study.

Reviewer 2 Report

The truth is I have mixed feelings about this manuscript, I found it potentially interesting, but the reviewed version has changed almost nothing despite giving the impression that the authors addressed my previous concerns. Here I explain the reasons why I am not convinced by the response of the authors:

1-    Flow diagram: the authors did not include information on the number of patients that did not qualify for the study for each of the criteria mentioned in the text.

2-    mRS: I think my previous comment was self-explanatory: please explain how the authors measured whether a patient had “No significant disability despite symptoms; able to carry out all usual duties and activities” during admission. Did they allow the patients to leave the hospital to check whether they could do their previous work duties without any difficulties?

3-    I do not fully understand whether the  authors used these consensus criteria and rewritten the text in the methods section to better reflect them or whether they reviewed all the cases in order to reassign this particular endpoint. I assume that the former is true because I don’t see any changes in the number of endpoints in the tables. 

4-    The new sentence “Butthe correlation between nasogastric tube placement and BRS remains to be verified by further comprehensive study because thepotential confounders such as stroke severity, stroke localization, consciousness level and so no.” seems to have several mistakes (underlined).

5-    The authors have only made very minor changes to this section, reducing it in 100 words (8%). I still think it is not focused enough.

Author Response

Responses to comments from Reviewer 2:

 We appreciate the constructive comments by Reviewer 2 and believe that quality of the second revised manuscript has improved by incorporating those comments. The followings are our responses to the specific issues raised:

 All the revisions are marked in the revised manuscript in red-color.
1) Flow diagram: the authors did not include information on the number of patients that did not qualify for the study for each of the criteria mentioned in the text.

Response: The authors apologize for the careless and have added the detailed information in the revised flow diagram on the number of patients that were not included in the study based on the exclusion criteria. (P. 2)

2) mRS: I think my previous comment was self-explanatory: please explain how the authors measured whether a patient had “No significant disability despite symptoms; able to carry out all usual duties and activities” during admission. Did they allow the patients to leave the hospital to check whether they could do their previous work duties without any difficulties?

Response: We thank the Review for bringing our attention to this point. Our team member visited the first ever stroke patient and evaluated patient’s disability degree according to the modified Rankin Scale (mRS) sheet. If the patients who had no significant disability despite symptoms we did not allow the patients to leave the hospital, instead the patients were hospitalized and mRS was checked until discharge. Nonetheless, per your original suggestion, we have deleted mRS (p. 5-6, Table 2; and p. 7, Figure 2) from the second revised manuscript.

3) I do not fully understand whether the authors used these consensus criteria and rewritten the text in the methods section to better reflect them or whether they reviewed all the cases in order to reassign this particular endpoint. I assume that the former is true because I don’t see any changes in the number of endpoints in the tables.

Response: Thank you for the comment. Following your comment, we have edited the statement to clarify that the criteria narrated in the manuscript are to better reflect the symptoms of the enrolled patients (p. 4, lines 122-124. The post-stroke pneumonia and UTI are defined according to CDC/NHSN surveillance definition of health care-associated infection and criteria in 2008 [1]. (p. 4, lines 124-140)

 4) The new sentence “But the correlation between nasogastric tube placement and BRS remains to be verified by further comprehensive study because the potential confounders such as stroke severity, stroke localization, consciousness level and so no.” seems to have several mistakes (underlined).

Response: We thank the Reviewer for this comment. Per suggestion in the original comment, we have deleted all the descriptions and citations of nasogastric tube and Foley in the second revised manuscript.

5) The authors have only made very minor changes to this section, reducing it in 100 words (8%). I still think it is not focused enough.

Response: Thank you for the comments. Per the instruction, we have further revised the Discussion section to remove the discussions on nasogastric tube and Foley, and shorten the session to 68%. We also added some new narrations to the Discussion per comments by the Reviewers (p. 8, lines 256-259; p. 9, lines 277-279 and 283-285).

  Reference:

1.         Horan, T.C.; Andrus, M.; Dudeck, M.A. Cdc/nhsn surveillance definition of health care-associated infection and criteria for specific types of infections in the acute care setting. American journal of infection control 2008, 36, 309-332.

  Once again, we thank the Reviewer for your insightful review and constructive comments to improve on the quality of the present study.
